# X-ray transient absorption reveals the $^1A_u$ (nπ⋆) state of pyrazine in electronic relaxation

Valeriu Scutelnic [1], Shota Tsuru [2,6], Mátyás Pápai [2,7], Zheyue Yang[1,8], Michael Epshtein[1,3,9], Tian Xue[1], Eric Haugen [1,3], Yuki Kobayashi[1,10], Anna I. Krylov [4], Klaus B. Møller [2], Sonia Coriani [2] & Stephen R. Leone [1,3,5✉]

Electronic relaxation in organic chromophores often proceeds via states not directly accessible by photoexcitation. We report on the photoinduced dynamics of pyrazine that involves such states, excited by a 267 nm laser and probed with X-ray transient absorption spectroscopy in a table-top setup. In addition to the previously characterized $^1B_{2u}$ (ππ⋆) ($S_2$) and $^1B_{3u}$ (nπ⋆) ($S_1$) states, the participation of the optically dark $^1A_u$ (nπ⋆) state is assigned by a combination of experimental X-ray core-to-valence spectroscopy, electronic structure calculations, nonadiabatic dynamics simulations, and X-ray spectral computations. Despite $^1A_u$ (nπ⋆) and $^1B_{3u}$ (nπ⋆) states having similar energies at relaxed geometry, their X-ray absorption spectra differ largely in transition energy and oscillator strength. The $^1A_u$ (nπ⋆) state is populated in 200 ± 50 femtoseconds after electronic excitation and plays a key role in the relaxation of pyrazine to the ground state.

[1] Department of Chemistry, University of California, Berkeley, CA, USA. [2] DTU Chemistry, Technical University of Denmark, Kongens Lyngby, Denmark. [3] Chemical Sciences Division, Lawrence Berkeley National Laboratory, Berkeley, CA, USA. [4] Department of Chemistry, University of Southern California, Los Angeles, CA, USA. [5] Department of Physics, University of California, Berkeley, CA, USA. [6]Present address: Ruhr-Universität, Bochum, Germany. [7]Present address: Wigner Research Centre for Physics, Budapest, Hungary. [8]Present address: Shanghai, China. [9]Present address: Beer-Sheva, Israel. [10]Present address: Stanford PULSE Institute, SLAC National Accelerator Laboratory, Menlo Park, CA, USA. ✉email: srl@berkeley.edu

Radiationless relaxation is crucial in a range of photochemical processes, such as vision[1], bioimaging[2], and photosynthesis[3,4]. By dissipating potentially harmful electronic energy into heat, radiationless relaxation provides protection of biological systems from sunlight[5,6]. Representative is the fast electronic relaxation through intermediate states responsible for the high photostability of nucleobases[7–10]. Despite its fundamental and practical significance, a complete mechanistic understanding of photoinduced transformations remains elusive. A particular example in this sense is the prototypical organic chromophore, the pyrazine molecule.

Excited states of pyrazine have been extensively investigated both experimentally[11–15] and theoretically[16–27]. Time-resolved photoelectron spectroscopy determined that pyrazine being promoted to the second dipole-allowed electronic state S$_2$—a $^1$B$_{2u}$ [$(1b_{1g})^{-1}(2b_{3u})^{+1}$] state of $\pi\pi^*$ character—converts in ~22 fs to its first excited singlet state S$_1$—a $^1$B$_{3u}$ [$(6a_g)^{-1}(2b_{3u})^{+1}$] state of $n\pi^*$ character—and then decays to the ground state S$_0$ ($^1$A$_g$) on a picosecond timescale[11,14,15] (see Fig. 1). The ultrafast $^1$B$_{2u}$→$^1$B$_{3u}$ internal conversion is driven by strong vibronic (i.e., non-adiabatic) coupling between the two states. However, a theoretical study by Werner and coworkers[19] set forth the hypothesis that an additional dipole-forbidden $^1$A$_u$($n\pi^*$) [$(6a_g)^{-1}(1a_u)^{+1}$] state is involved in the photoinduced dynamics of pyrazine on the 10 fs to sub-picosecond timescale. The possible role of the $^1$A$_u$ state was further investigated in two consecutive theoretical studies by Sala and coworkers[20,21], who predicted a rapid increase of population in $^1$A$_u$ as well as in $^1$B$_{3u}$ during the fast decay from $^1$B$_{2u}$. A theoretical work authored by some of us[25] attributed the fast decay from the $^1$B$_{2u}$ state to vibronic couplings with both the $^1$B$_{3u}$ and the $^1$A$_u$ states and predicted oscillatory population dynamics that was explained by the periodic switching of the energetic ordering of the $^1$B$_{3u}$ and the $^1$A$_u$ states driven by vibrations within the aromatic ring. Direct experimental confirmation of this dynamics has been hindered so far by the short timescale of the process and the overlap of signals from different electronic states.

Arguments against the participation of the $^1$A$_u$ state in the fast non-radiative decay of pyrazine have also been presented. Kanno et al.[22] concluded that the probability of transitions to the $^1$A$_u$ state with vibronic coupling is so low that a two-state picture provides an adequate representation of the relaxation dynamics. A time-resolved photoelectron imaging (TR-PEI) study by Horio et al.[15] did not find evidence supporting the involvement of the

$^1$A$_u$ state. Mignolet et al.[23], who later simulated the TR-PEI spectra, supported the conclusions of Kanno et al. and Horio et al.

The controversy regarding the involvement of the $^1$A$_u$ state persists because, on the one hand, it is difficult to experimentally disentangle individual contributions of various electronic states involved in the photoinduced dynamics and, on the other hand, the energy ordering of the states depends on the level of theory used in the electronic structure calculations. Therefore, alternative time-resolved experimental techniques capable of tracking the photoinduced dynamics of pyrazine and separating the contributions of the individual excited states are highly desirable, along with additional theoretical simulations. Indeed, according to the theoretical study by Sun et al.[27] it should be possible to identify spectroscopic signatures of the $^1$A$_u$ state with three techniques: time- and frequency-resolved fluorescence spectroscopy, electronic two-dimensional spectroscopy, and transient absorption pump-probe spectroscopy.

X-ray absorption spectroscopy opens new horizons in observing elusive intermediates of photoexcited molecules on femtosecond timescales[7,8,28–31]. X-rays deliver a unique element specificity because they involve the transitions from localized core orbitals, whose energies in different types of atoms are often separated by hundreds of electron volts. Moreover, core-level transitions are sensitive to the local chemical environment and can report on shifts in electron density in the proximities of the probed atoms[28,32–34]. With recent advances in high-order harmonic generation (HHG), water-window X-ray table-top setups for transient absorption are available for solving photochemistry problems[30,35].

In this work, we investigate the photoexcited dynamics of pyrazine by means of ultrafast soft X-ray transient absorption spectroscopy. The X-ray spectrum, which covers the entire region of the carbon K-edge signals (~280 eV), is produced with a table-top setup via HHG. With this recently developed light source, the time-resolved internal conversion chain of $^1$B$_{2u}$→$^1$B$_{3u}$→⋯→$^1$A$_g$ is monitored revealing dramatic variations in the molecular geometry and electronic configurations. To interpret the experimental spectra, we compute X-ray absorption spectra from carbon $1s$ orbitals for pyrazine in different electronic states (the ground state, $^1$B$_{2u}$, $^1$B$_{3u}$, and $^1$A$_u$ states) and carry out non-adiabatic nuclear dynamics simulations. The evidence obtained from experiment and theory points towards the involvement of the dipole-forbidden $^1$A$_u$ state, which becomes populated within the first (200 ± 50) fs.

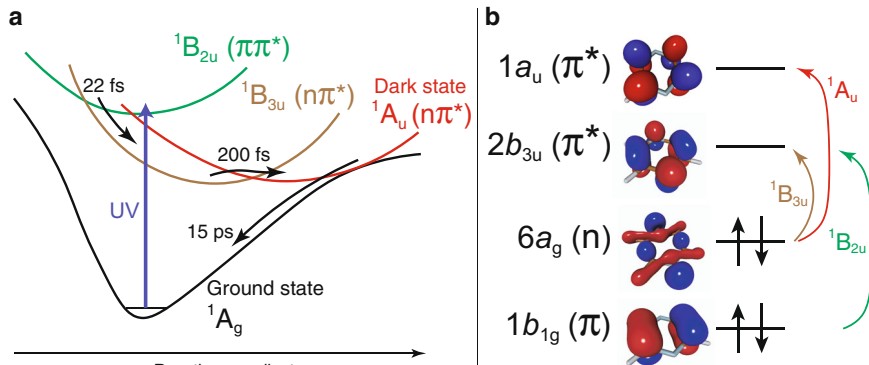

**Fig. 1 Potential energy surfaces and molecular orbitals of pyrazine.** a Upon absorption of an ultraviolet photon (blue vertical arrow) energy flows through the $^1$B$_{2u}$ (green), $^1$B$_{3u}$ (brown), and $^1$A$_u$ (red) states; timescales in the sequence $^1$B$_{2u}$→$^1$B$_{3u}$→$^1$A$_u$→$^1$A$_g$ are represented with curved arrows. b Arrows represent dominant transition characters of the valence excited states, relevant molecular orbitals are shown with their irreducible representations of the $D_{2h}$ point group. Supplementary Table 1 provides the analysis in terms of natural transition orbitals[46,47].

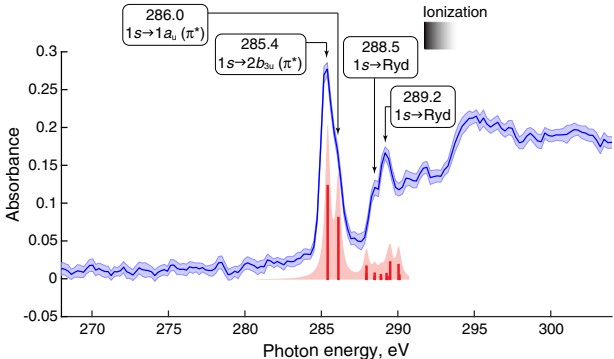

**Fig. 2 Experimental and theoretical pyrazine absorption spectra in the ground electronic state.** Experimental absorption spectrum (blue line). The shaded area covers 95% level of confidence (two standard deviations of 32 measurements). Numbers indicate the absorption peak positions in eV. The red bars and pink area correspond to the computed spectra (B3LYP/6-311+ +G**, shifted by +10.7 eV). Black shaded area shows the ionization threshold.

## Results

**Ground-state spectroscopy.** Fig. 2 shows the near edge X-ray absorption fine structure (NEXAFS) spectra of ground-state pyrazine obtained in the present study. The experimental spectrum is in good agreement with previous data obtained at a synchrotron facility[36]. The computed spectrum of the ground state is blue-shifted by 10.7 eV to align it with the experimental blue line. Natural transition orbitals (NTOs) of the bands indicated by arrows are given in Supplementary Table 2. Because NTOs in pyrazine closely resemble the canonical molecular orbitals (Fig. 1b), the latter is used in the following assignment. The band peaking at 285.4 eV and the shoulder at 286.0 eV are assigned to the $1s\rightarrow2b_{3u}$ ($\pi^*$) and $1s\rightarrow1a_u$ ($\pi^*$) transitions, respectively (see orbital nomenclature in Fig. 1b). The absorption feature at 288.5 eV has been previously assigned to a transition from $1s$ to $\sigma^*$ symmetry orbital and the band at 289.2 eV to two $1s$ transitions to orbitals of $\sigma^*$ and $\pi^*$ character[36]. Here, the band at 288.5 eV is found to have Rydberg character (NTOs in Supplementary Table 2) possibly mixed with hydrogen-derived $\sigma^*$ antibonding orbitals. The band at 289.2 eV has Rydberg character with no $\sigma^*$ component. The NTOs and the corresponding assignments are in line with those obtained at the coupled-cluster singles and doubles level of theory[25]. With the good agreement between the experiment and theory reached for the ground state spectrum, we proceed to analyze excited-state spectra.

**Time-resolved measurement.** The photoinduced dynamics of pyrazine is reported in Fig. 3 via differential absorbance, which is the change in X-ray absorption with and without the 267 nm excitation pulse. The time delay between pump and probe pulses was uniformly varied from −40 to 220 fs in 20 fs steps. The negative absorption signals at 285.4 and 289.2 eV align with the strong absorption bands of the ground state and are thus assigned to depletion of the ground-state population. In the first few tens of femtoseconds, a positive absorption signal is prominent at 281.5 eV. This feature evolves for ~100 fs and is replaced with a broad absorption covering 282–284 eV. At the same time, a notably more intense absorption band at 284.5 eV appears, slightly delayed compared with the emergence of the other time-transients, along with another intense band at 288.0 eV. To identify the timescales of the photoinduced dynamics, we plot in Fig. 4 the differential absorbance at several energies, marked with arrows in Fig. 3, as a function of time delay. The delay-dependent signals are fitted with a convolution of the instrument response function (IRF) and the corresponding exponential decay or growth function. Following

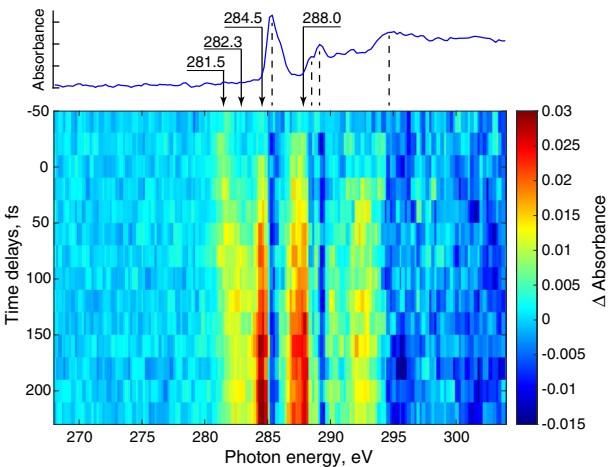

**Fig. 3 Time-dependent differential absorbance.** A two-dimensional map represents the differential absorbance with dependence on X-ray photon energy and the time delay between 267 nm pump and probe pulses. The top blue curve is the X-ray absorption spectrum of the ground-state pyrazine (the same plot as in Fig. 2) and vertical dashed lines project the ground-state absorption bands onto the negative (bleach) signals of the 2D map. Arrows point to absorbance energies specific to different excited states. A similar map at long time delays is given in Supplementary Fig. 1.

this analysis, three-time constants were determined: ~90 fs decay, ~50 fs growth, and ~200 fs growth for the transients at 281.5 eV, 282.3 eV, and the 284.5 (and 288.0) eV, respectively.

**Spectral assignment.** To further assign the highlighted bands and attribute the timescales, we calculated the NEXAFS spectra for the excited states, which are averaged over a distribution of structures that arises from the zero-point vibrational energy (Wigner distribution) for the ground state of pyrazine. First, we compare the experimental spectrum at −20 fs delay with the absorption spectrum of the $^1B_{2u}$ state in Fig. 5a, and the experimental spectrum at 20 fs delay with the absorption spectrum of the $^1B_{3u}$ state in Fig. 5b. Supplementary Note 5 discusses geometrical effects on the low-energy $^1B_{2u}$ and $^1B_{3u}$ peaks. Relevant NTOs at the Franck-Condon (FC) geometry are shown in Supplementary Tables 3 and 4, respectively.

First, the lowest-energy feature (Fig. 4a) is assigned to the $1s\rightarrow1b_{1g}$ ($\pi$) transition from the $^1B_{2u}$ state based on the comparison in Fig. 5a. Exponential fitting of the experimental data in Fig. 4a yields a decay lifetime of (90 ± 70) fs; the large uncertainty is affected by the low signal-to-noise of this transient. Even though the limited time resolution of the present experiment precludes a precise measurement of the lifetime of the $^1B_{2u}$ state, the data are compatible with the ~22 fs relaxation time determined by photoelectron spectroscopy[13–15]. The over-estimated lifetime of $^1B_{2u}$ can be also caused by the weakly overlapping higher energy signal from another electronic state down to 281.5 eV.

Second, the absorption band ~282.3 eV, which exhibits a slight blue shift (~0.8 eV) from the $^1B_{2u}$ state feature at 281.5 eV, is assigned to the $1s\rightarrow6a_g$ (n) transition from the $^1B_{3u}$ state (see Fig. 5b). The (50 ± 30) fs growth time constant of $^1B_{3u}$ (Fig. 4b) is consistent with the expected ~22 fs internal conversion[13–15]. The blue shift from the $1s\rightarrow1b_{1g}$ ($\pi$) transition of the $^1B_{2u}$ state is justified by the higher energy of the $6a_g$ orbital than $1b_{1g}$ (orbital diagrams are shown in Fig. 5a, b). At longer delays, this band broadens, which we attribute to the excess vibrational energy of ≈0.8 eV available in the $^1B_{3u}$ state. Although the positions of the 281.5 eV and 282.3 eV low-energy peaks are underestimated by

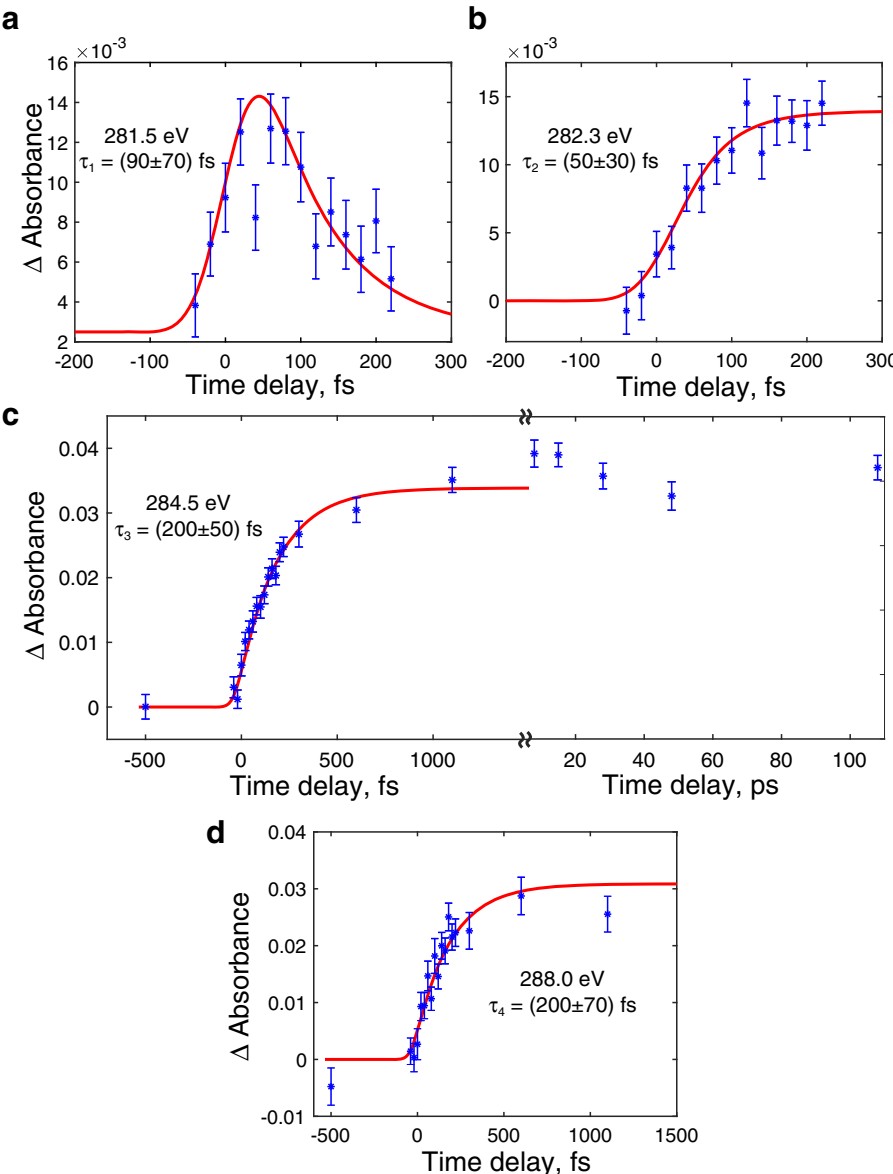

**Fig. 4 Time dependence of absorbance at selected energies. a** 281.5 eV, red line is the convolution of the instrument response function (IRF) with exponent $e^{-t/\tau_1}$, where $\tau_1 = (90 \pm 70)$ fs. **b** 282.3 eV, red line is the convolution of IRF with the function $(1 - e^{-t/\tau_2})$, where $\tau_2 = (50 \pm 30)$ fs. **c** 284.5 eV, red line is the convolution of IRF with $1 - e^{-t/\tau_3}$, where $\tau_3 = (200 \pm 50)$ fs. **d** 288.0 eV, red line is the convolution of the IRF with $1 - e^{-t/\tau_4}$, where $\tau_4 = (200 \pm 70)$ fs. Error bars in all panels represent one standard deviation of 128 measurements.

the calculations, by 0.5 eV and 0.4 eV, respectively, the experimentally observed 0.8 eV blue shift is well reproduced (0.9 eV).

Crucially, the spectrum at 20 fs delay exhibits a shoulder at 284.5 eV (marked with an asterisk in Fig. 5b), red-shifted from the ground-state main peak by 0.9 eV. This red-shifted peak is not reproduced in the simulated absorption spectrum from the $^1B_{3u}$ state (the $^1B_{3u}$ main peak is blue-shifted). The signal at 284.5 eV grows with a time constant of $(200 \pm 50)$ fs (see Fig. 4c), distinctly different from the one at 282.3 eV rising with the time constant of $(50 \pm 30)$ fs (Fig. 4b). Furthermore, the asymptote intensity of the signal at 284.5 eV is two times larger than the maximum at 282.3 eV. To the best of our knowledge, the 200-fs rise-time constant was not identified in previous experimental studies of pyrazine. At 220 fs (the longest delay depicted in Fig. 3) the feature at 284.5 eV dominates the spectrum (Fig. 5c). By this time population transfer to the optically dark $^1A_u$ state is expected, and the 284.5 eV feature, which cannot be explained by the $^1B_{2u}$ (Fig. 5a) or $^1B_{3u}$ (Fig. 5b) state, is suggestive of its contribution.

The experimental band at 284.5 eV aligns well with the modeled absorption signal at 284.85 eV for the Wigner distribution of the $^1A_u$ state (see the green curve of Fig. 5c and NTOs at the FC geometry in Supplementary Table 5). The computed peak at 284.85 eV is assigned to $1s \rightarrow 2b_{3u}$ ($\pi^*$) transitions and is 0.55 eV red-shifted with respect to the same $1s \rightarrow 2b_{3u}$ ($\pi^*$) transition in the absorption spectrum of the ground state peaking at 285.4 eV (Fig. 2). For interpreting this large redshift we analyze the orbitals involved in core-valence excitations of the $^1A_u$ and ground states. Since the $1a_u$ orbital is strongly localized on carbon atoms, the core-excited state reached from $^1A_u$ is stabilized due to reduced Coulomb repulsion between the electrons occupying the $1a_u$ and $1s$ orbitals and a more effective attraction between the electron in the $1a_u$ orbital and carbon nuclei (see Supplementary Fig. 4). A slightly delayed increase of the population in the $^1A_u$ state during the ultrafast relaxation (Fig. 4c) is qualitatively consistent with the nuclear dynamics simulation of Sala et al.[21]. The substantial discrepancy (0.9 eV, shown with a purple line in Fig. 5c) between the calculated $1s \rightarrow 6a_g$

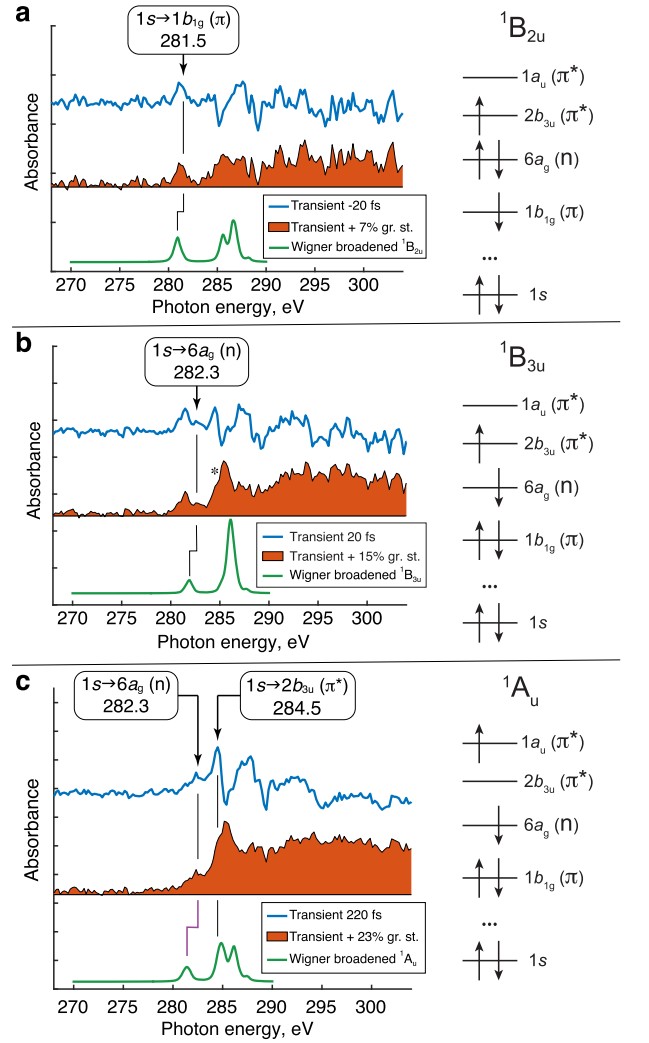

**Fig. 5 Experimental differential spectra acquired at different time delays.** **a** −20 fs delay (blue), and the computed $^1B_{2u}$ spectrum (green). **b** 20 fs delay (blue), and the computed $^1B_{3u}$ spectrum (green). **c** 220 fs delay (blue), and the computed $^1A_u$ spectrum (green). All the computed spectra are calculated for the ground-state Wigner distribution. The brown-filled spectra are the differential traces corrected for ground-state bleach. The added ground state percent is determined from the pump fluence and absorption cross-section of pyrazine at 267 nm; for −20 and 20 fs delays the temporal overlap of the pump and probe is also taken into account, see Supplementary Note 3. Prevailing electron configurations of the corresponding valence excited states are sketched on the right.

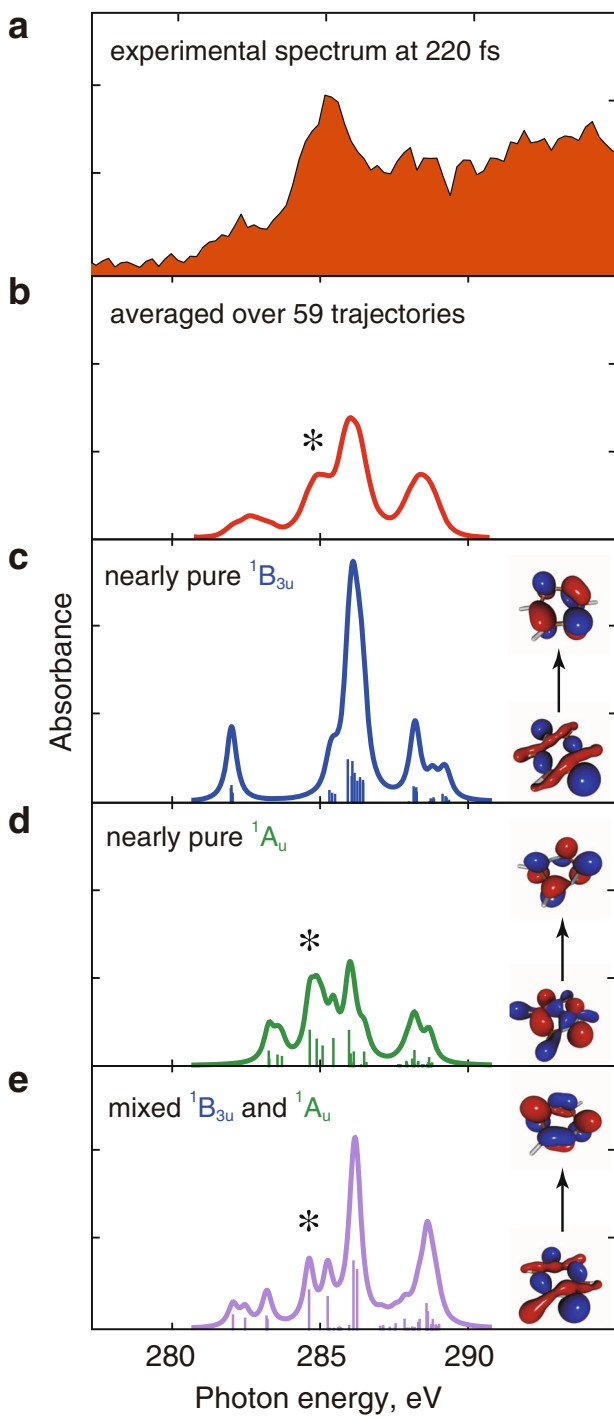

**Fig. 6 NEXAFS spectra of selected geometries from the surface hopping simulation.** **a** Brown area is the differential spectrum at 220 fs corrected for ground-state bleach (see Fig. 5c). **b** Red curve is the spectrum averaged over all the geometries of the 59 trajectories of the FSSH simulation at 220 fs. **c** Computed absorption spectrum from nearly pure $^1B_{3u}$ state. **d** Computed absorption spectrum from nearly pure $^1A_u$ state. **e** Computed absorption spectrum from a mixed $^1B_{3u}$ and $^1A_u$ state. Blue, green, and purple lines are calculated for different geometries extracted from the trajectories at 220 fs. A natural transition orbital pair in each panel shows the dominant transition character at the chosen molecular geometry. Asterisks indicate the transitions characteristic of the $^1A_u$ state.

(n) transition at 281.4 eV and the lowest-energy experimental feature peaking at 282.3 eV is rationalized by the nuclear motion—in the first 220 fs the wave packet will have departed from the FC vicinity.

For this reason, we investigate the implication of the nuclear motion on the X-ray resonances by calculating the spectra of the excited states averaged over the 59 trajectories from the fewest-switches surface hopping (FSSH) simulation at 220 fs (red curve in Fig. 6b)—at this time point, only two trajectories remain in the $^1B_{2u}$ state and the other 57 trajectories are in the lowest excited state having a mixed $^1B_{3u}/^1A_u$ character. The computed averaged spectrum (red trace) exhibits a very good agreement with the experimental one (brown area in Fig. 6a, same as that in Fig. 5c) in the X-ray region below ionization energy.

To map the spectral features onto the electronic configurations of the underlying states, we plot 3 out of the 59 spectra used to construct the averaged red spectrum of Fig. 6b. In Fig. 6c, the lowest excited state maintains the configuration character of an almost pure $^1B_{3u}$ state, Fig. 6d corresponds to a nearly pure $^1A_u$ state, and Fig. 6e features a mixed $^1B_{3u}/^1A_u$ character. The peaks between 284.5 and 285.0 eV (marked with asterisks in Fig. 6b, d, and e) appear only when the $^1A_u$ configuration is involved. This assignment is further supported by Supplementary Figs. 6 and 7 that evidence negligible contribution from $^1B_{3u}$ in the 284.5 −285.0 eV region upon nuclear relaxation. The remaining 56 spectra show analogous characteristics, apart from the two trajectories that remain in the $^1B_{2u}$ state. Therefore, we attribute the 284.5 eV experimental absorption band (Fig. 5c) to the optically dark $^1A_u$ state. From the relative intensities of the peaks ~284.5 and 286.0 eV of the averaged spectrum (red trace in Fig. 6b) and the calculated Wigner broadened spectra for the $^1B_{3u}$ and the $^1A_u$ states (green traces in Fig. 5b, c, respectively), we estimate the population of the $^1A_u$ state at 220 fs to be ~60%. This ratio is also in agreement with the theoretical prediction of Sala et al. (~50%)[21]. A complementary decaying component with a 200 fs timescale is not noticeable in any of the spectral data shown in Fig. 3 due to the overlap of the peaks with $1s{\rightarrow}6a_g$ (n) transition character from both the $^1B_{3u}$ and $^1A_u$ states at around 282.3 eV, and also owing to the interference of the intense peak at 286.1 eV in the calculated spectrum for the $^1B_{3u}$ state (see Supplementary Table 4) with the ground-state bleach (see Fig. 3).

The intensity of the experimental band at 288.0 eV increases on a timescale similar to that of the 284.5 eV peak (compare Fig. 4c, d), suggesting that the absorption signal at 288.0 eV is also dominated by the $^1A_u$ state, even though this band appears in the computed spectra of all three excited states (Figs. 5 and 6). We assign this absorption band to a transition from $1s$ to Rydberg orbitals (see Supplementary Tables 3–5), similar to the one at 288.5 eV in the spectrum of the ground state (Fig. 2).

On picosecond timescales, the absorption at 282.3 eV decreases (Supplementary Fig. 9). The low signal-to-noise prevents the determination of the corresponding lifetime, nonetheless, the data points suggest a decay, consistent with the internal conversion to the ground state with a 15–20 ps time constant[15]. Consequently, once the equilibrium between $^1B_{3u}$ and $^1A_u$ is reached in ~200 fs, internal conversion to the ground state can occur from both states. The spectrum acquired at 28 ps time delay corrected for the ground-state bleach (brown area in Supplementary Fig. 11) strongly resembles the shape of the broadened ground state (red trace in Supplementary Fig. 11). A similar broadening has been previously observed in vibrationally hot cyclohexadiene[30]. The band that appears at 284.5 eV (assigned to the $^1A_u$ state) overlaps with the ground-state absorption, giving a reason for the constant intensity at 284.5 eV after the internal conversion to the ground state. However, the internal conversion to the hot ground state in the first 200 fs is excluded (see Supplementary Note 9).

Finally, we revisit the TR-PEI experiment[15] where the 0–1.5 eV and 3–4 eV bands appear on sub-picosecond timescales, and they are both assigned by the authors to ionization processes from the $^1B_{3u}$ state. The 3–4 eV band stems from ionization into the $D_0(^2A_g)$ state of the pyrazine cation, which can also be reached from the $^1A_u$ state. Our calculations suggest that the $^1B_{3u}$ and $^1A_u$ states are nearly isoenergetic at their relaxed geometries, therefore these states might be indistinguishable by photoelectron kinetic energy.

## Discussion
We have applied ultrafast soft X-ray transient absorption spectroscopy to probe the excited-state dynamics of pyrazine. Short-

time dynamics distinguish between the $^1B_{2u}$ and $^1B_{3u}$ states, and a distinct transient signal with $(200 \pm 50)$ fs time constant is identified and assigned here to the $^1A_u$ state, which is optically inaccessible from the ground state. The evidence in the present work supports the substantial role of the $^1A_u$ state in the photoinduced dynamics of pyrazine, as suggested by previous theoretical studies. Carbon K-edge transient absorption spectroscopy with higher time resolution will be the subject of future studies.

## Methods
**Experimental details**. A detailed description of the transient carbon K-edge spectrometer is available elsewhere[29,30]. In brief, 11.5 mJ pulses of a Ti:sapphire amplifier (1 kHz repetition rate, 35 fs pulse duration) are split 10:90 for pump and probe generation, respectively. The low power arm is frequency tripled in β-barium borate crystals to produce 267 nm[37] to pump the pyrazine molecules, and the higher power arm is converted in a HE-TOPAS optical parametric amplifier to 1470 nm. The 2.5 mJ per pulse mid-IR pulses are focused with an $f = 30$ cm lens into a helium gas target (flowing at 1500 Torr) to drive HHG. A 100 nm thick aluminum filter transmits the high harmonics and blocks the intense infrared light. A toroidal mirror focuses the high harmonics into a flow cell (300 μm holes for laser entrance and exit) filled with the vapor of pyrazine (purchased from Sigma-Aldrich, 99% purity) at a temperature of 320 K. In the cell, the X-rays interrogate pyrazine molecules via a controlled delay after 267 nm excitation. The 11 μJ UV pulses, being focused with an $f = 45$ cm lens, provide a ~$2.5 \times 10^{11}$ W cm$^{-2}$ peak pump intensity and excite ~25% of the pyrazine molecules in the pump volume. X-rays that pass through the flow cell are dispersed by a grating and measured on a charge-coupled device X-ray camera. Photons of the same energy arrive at the same horizontal location on the camera and using argon[38] and allyl radical[39] transition lines for wavelength calibration we reconstruct the X-ray spectrum. Each image is accumulated for 1 s.

In absorption spectra, absorbance $A$ is defined as $A = -\log_{10}\frac{I}{I_0}$, where $I$ and $I_0$ are the intensity of transmitted X-rays through the interaction region with and without pyrazine molecules, respectively. The differential absorbance $\Delta A$ is defined as $\Delta A = -\log_{10}\frac{I_{withUV}}{I_{withoutUV}}$, where $I_{with\ UV}$ and $I_{without\ UV}$ are the intensity of transmitted X-rays with and without UV pump pulses, respectively.

The one-photon absorption of the UV pump is confirmed in a power-dependence study (see Supplementary Figs. 12, 13).

The AC-Stark shift of argon core-excited Rydberg states allows us to determine the cross-correlation of pump and probe pulses as being $(82 \pm 13)$ fs (Supplementary Fig. 2).

**Computational details**. The ground-state geometry of pyrazine is optimized using DFT with the B3LYP functional[40] and the aug-cc-pVTZ basis set[41,42]. Excitation energies and oscillator strengths of the core-to-valence transitions from the ground state and from the three lowest valence singlet states are calculated at the TDDFT/B3LYP/6-311++G** level of theory. The core-valence separation scheme is used to access the core-excited states[43] and the maximum overlap method[44] is employed to obtain reference wave functions corresponding to the valence excited states. The X-ray absorption spectra are simulated by Lorentzian broadening (FWHM = 0.4 eV) of the calculated stick spectra and by shifting the theoretical values by +10.7 eV before comparison with the experimental data. The +10.7 eV constant shift is a systematic error of TDDFT, attributable to relaxation of the electrons caused by the core-hole and by self-interaction errors and is determined as the value needed to align the calculated ground-state spectrum with the corresponding experimental one. All electronic structure calculations are performed with the Q-Chem 5.3 electronic structure package[45]. NTOs[46,47] serve as a guide in the analysis and assignment of the transitions. This analysis is particularly important for structures different from the FC geometry.

In the dynamics simulations, we employ the FSSH method as implemented in SHARC 2.1[48,49], with TDDFT/B3LYP/cc-pVDZ as the electronic structure method, which is provided by the ORCA[50] backend. To sample initial conditions, we generate 200 pairs of coordinates and momenta according to the Wigner distribution[51] in the ground state. These initial conditions are used to carry out the FSSH simulations for the 200 pairs using a UV laser field at resonance with the $^1B_{2u}$ state that is enveloped by a Gaussian of 1.0 TW cm$^{-2}$ intensity and 80 fs FWHM, up to the delay time of 340 fs (time zero is considered as the instant of the laser pulse maximum). An energy-based decoherence scheme is applied to the diagonal states[52]. A standard (non-relativistic) electronic Hamiltonian, called molecular Coulomb Hamiltonian, without the inclusion of spin-orbit coupling (SOC) is utilized. The wave-function coefficients are propagated by the local diabatization technique using wave-function overlaps[53]. By choosing the laser intensity of 1.0 TW cm$^{-2}$, 59 of the 200 trajectories (29.5%, close to the value of ~25% in the experiment) undergo excitation from the ground state to the $^1B_{2u}$ state. The FSSH simulation is implemented using the adiabatic electronic states, whereas the results are discussed in terms of the diabatic electronic states.

In order to compute the X-ray absorption spectrum at a 220 fs delay, chosen to match key features in a well-characterized longer time delay in the experimental data, the molecular geometries are collected from 59 trajectories that have undergone an excitation to the $^1B_{2u}$ state. The X-ray absorption spectrum of the ensemble at 220 fs is obtained as the average of the Lorentzian-broadened spectra calculated for the 59 molecular geometries. The convergence has been tested by computing the trajectory-averaged spectrum with 10, 20, 30, 40, and 50 of the 59 trajectories, thereby confirming that the surface hopping simulation employing 59 trajectories has converged (see Supplementary Fig. 8).

To obtain the geometries of the molecular ensemble after radiationless decay into the ground state, we have performed additional FSSH calculations including SOC[54]. Details of these calculations are given in Supplementary Note 8.

## Data availability

The experimental data and the input/output of the FSSH simulations are available in the Zenodo database [https://doi.org/10.5281/zenodo.5077821].

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

## Acknowledgements

V.S., M.E., T.X., E.H., Z.Y., and S.R.L. gratefully acknowledge the generous support from the U.S. Department of Energy, Office of Science, Office of Basic Energy Sciences (Contract no. DEAC02-05CH11231), the gas phase chemical physics program through the Chemical Sciences Division of Lawrence Berkeley National Laboratory. V.S. acknowledges support from the Swiss National Science Foundation (P2ELP2_184414). YK was supported by the National Science Foundation (CHE-1951317) and the Army Research Office (W911NF-14-1-0383). S.T. acknowledges support from the European Union's Horizon 2020 research and innovation program under the Marie Skłodowska-Curie Grant Agreement no. 713683 (COFUNDfellowsDTU). M.P. acknowledges support from the Hungarian National Research, Development and Innovation Fund under Grant no. NKFIH PD 134976. S.C., M.P., and K.B.M. acknowledge the Danish Council for Independent Research (now Independent Research Fund Denmark), Grant nos. 7014-00258B (S.C.), 4002-00272 (M.P., K.B.M.), and 8021-00347B (M.P., K.B.M.). A.I.K. was supported by the U.S. National Science Foundation (no. CHE-1856342).

## Author contributions

V.S., Z.Y., M.E., and S.R.L. designed the experiment. V.S., M.E., T.X., E.H. collected the data. S.T., M.P., A.I.K., K.B.M., and S.C. developed the theoretical model and defined the computational approach. S.T. and M.P. carried out the calculations. V.S., S.T., M.P., Y.K., K.B.M., S.C., and S.R.L. interpreted the results and wrote the manuscript. All authors discussed the science and the article.

## Competing interests

The authors declare the following competing interests: A.I.K. is the president and a part-owner of Q-Chem, Inc. The other authors declare no competing interest.
