## [Peer Review File · Nature Communications]

Reviewers' comments:

Reviewer #1 (Remarks to the Author):

Scutelnic et al report on the experimental detection of the 1Au state in photoexcited pyrazine. Their method is based on ultrafast table-top HHG soft X-ray spectroscopy at the carbon K-edge with ~ 90 fs temporal resolution. The optically dark 1Au state has been predicted theoretically - this would be the first experimental evidence for its population. The authors compare their challenging experimental data with ab initio TDDFT and FSSH calculations of the excited states and corresponding oscillator strengths for the core-level transitions originating in the 1s level.

Overall, this is a well-written paper and the results are presented in a clear way. The results are of high impact in the physical chemistry community. However, unfortunately, the S/N ratio in the data is lacking and my overall impression is that the assignment to the 1Au state (or a mixture of 1Au and 1B3u) is not very convincing. One particular point of concern is the fact that the authors had to add a varying fraction of the ground-state spectrum to reconstruct the excited state NEXAFS spectra. They state that the fraction is changing over time due to a change in spatial overlap. I understand these are extremely challenging experiments, but a time-varying spatial overlap between pump and probe is really worrying.... The authors do not adequately address this problematic. From their description in the SI, it is not clear to me what criteria are applied to determine the fraction of GS spectrum that is added. Within the S/N level of the data, a range of fractions would still yield reasonable spectra (with reasonable meaning that cross sections are not negative, nor sharp peaks appear in an otherwise smoothly varying spectrum). It is not clear to me to which extent the uncertainty in ground-state contribution affects the shape of the excited-state spectrum at different time delays (and how this in turn may affect the spectral assignments and kinetic fitting).

I also have some concerns about the reliability of the fitted time constants given the rather large IRF and uneven sampling along the time axis. The authors state that the fitted decay constant for the 1B1u state is "strongly affected by the IRF". What does this mean? Aren't the authors fixing the IRF in the fitting and properly weighting the fit with its respective error bar? Is there additional uncertainty expected in the IRF, i.e. variations as a function of (real) time? Also, the time stepping at later delay times is not the same as at early delay times, which may affect the statistical significance of the difference in reported decay constants τ_{1} and τ_{2} .

When I look at Fig. 5 and I compare the simulated green spectra with the GSB-corrected experimental orange spectra, I am just not convinced about the agreement... In particular, the lowest-energy pre-edge peak is really off for all time delays and it is not entirely clear why (does this mean the authors aligned the theoretical and experimental spectra based on the feature around 284 eV; if yes, how does this affect the assignment of the 1Au shoulder feature denoted with an asterix?). I think the authors mention the motion of the wave packet away from the FC region to be responsible for this discrepancy for the longest time delays (~ 220 fs). It's not clear how this affects the earlier times though.

In this context, a calculation of the ground-state spectrum is missing. This would be very useful to estimate the expected accuracy of the theory (if the GS is bad already, can we expect it to be as good/better for the ES?). Are core-hole effects in the GS and/or ES spectra expected to play a role and accounted for in the theory?

The simulation of the ~ 220 fs spectrum using FSSH, however, shows reasonable agreement with the experiment, which adds significantly to the credibility of the conclusions. The major issues with the GSB correction, however, still make this agreement less convincing than hoped for...

Given the very large laser fluences (30 mJ/cm) for solid state samples, the authors should carefully comment on their strategy to mitigate and check for sample damage.

In conclusion, given the concerns described above, I do not recommend publication in Nat Comm at this point. If the assignment of the 1Au state can be solidified by addressing these concerns, I agree that the impact would be large and publication in Nat Comm may be suitable.

Reviewer #2 (Remarks to the Author):

X-ray transient absorption reveals the 1Au (nn*) state of pyrazine in electronic relaxation

This manuscript reports the time-resolved X-ray absorption spectrum of pyrazine following femtosecond excitation at 267nm, thereby primarily preparing the optically bright $\pi\pi^*$ electronic state. The evolution of the electronic character of the prepared wave packet is monitored via the evolution of the X-ray absorption spectrum (XAS) and interpreted using surface hopping simulations and XAS simulated using ab initio electronic structure methods. The authors assign the delayed rise of a prominent spectral feature to the 1Au state, whose role in the electronic relaxation pathways of pyrazine has been the source of recent discussion in the literature. This work employs novel/emerging time-resolved spectroscopic techniques in conjunction with state of the art electronic structures approaches on a problem that highlights the potential of ultrafast X-ray spectroscopy to image photo-physical processes. Each of these elements make this work appropriate for publication in Nature Communications, but the key assignment of this work, that of the role of the 1Au state in the excited state dynamics, is not conclusive.

The first 3 comments speak directly to the assignment of specific spectral features to the Au state, while the remaining points address more general questions regarding the analysis of the spectrum.

1. The most significant issue concerns the high degree of overlap between the XAS from the B3u and Au states. As Figure 6 shows, given the resolution of the experiment, are these results also consistent with the the wave packet being exclusively B3u character? Furthermore, it is not clear, on the basis of Figures 5c and 6 alone, that the shoulder of the Au spectrum at 284.5 eV is consistent with the dominant spectral feature in the experimental TRXAS.
2. It does not appear that the peak at 284.5eV decays even after 200ps. The authors attribute this behavior to the rise in vibrationally excited ground electronic state population. While this assignment for the long time (> 20ps) is reasonable, without an observable decay how is it possible to separate excited signal from ground state signal? Therefore, the key peak used in the assignment overlaps not only with a different excited state, but also the ground state.
3. In Figure 5c, the peak at 284.5 is seemingly assigned to the a peak in the Au spectrum at the FC region. On the basis of the discussion, is this reasonable? The trajectory simulations appear to indicate that the structural relaxation results in a shift of the "Au" spectrum. The trajectory-averaged signal shows only a small shoulder at this energy.
4. The authors' analysis that yields a result 23% of the sample excited via a single photon process is quite high. It is difficult to excite more than 10% of a sample before multi-photon absorption processes occur. The fact that the IP of pyrazine is almost exactly $2 \times 4.67\text{eV}$ ($\sim 9.3\text{eV}$) adds to this concern. Did the authors confirm pyrazine cation is not contributing to the TRXAS?
5. Regarding Figure 5: why was the trajectory averaging procedure only performed for the 220fs

spectrum, this analysis would also be relevant for showing vibrational/nuclear dynamics on the B3u state?

6. A surface hopping simulation employing only 59 trajectories (with this many degrees of freedom and this many electronic states) is likely not converged (eg. wrt to electronic populations). While the authors don't lean on these results too hard, this would affect both the fraction/prevalence of Au state present in the simulation, but perhaps more importantly, how energetically broad the expected spectral signature would be at, for example 220fs.

In summary, this work is of high quality and employs powerful new ultrafast spectroscopy and high levels of ab initio computation to investigate a benchmark photophysical system. However, the results and analysis are unable to definitively assign the contribution of an elusive optically dark electronic state to the wave packet dynamics, in the opinion of this referee, which makes it more suitable for a more specialized journal.

Reviewer #3 (Remarks to the Author):

In this work, the femtosecond relaxation dynamics of the valence excited states of the pyrazine molecule in the gas phase was investigated with X-ray transient absorption (TA) pump-probe (PP) spectroscopy. While pyrazine as such is not a particularly exciting chromophore, a wealth of experimental and theoretical data on the femtosecond relaxation dynamics of the valence excited states of pyrazine are available. The interpretation of the signals of the relatively new X-ray TA PP technique can thus be scrutinized in comparison with well-established knowledge. It is very interesting that the time-dependent population of the Au(np π^*) state can be measured with X-ray TA PP spectroscopy, because the population dynamics of this dark state cannot be detected with conventional UV TA PP spectroscopy. The present work is of broad interest for the highly active field of ultrafast X-ray spectroscopy with novel light sources and is therefore recommended for publication in Nature Communications. A few issues should be clarified in a revision of the manuscript.

(1) The authors report a time constant of 200 fs for the transfer of population from the bright B2u(p π^*) state to the dark Au(np π^*) state. Quantum dynamics calculations for a nine-dimensional model derived from XMCQDPT2 ab initio calculations predicted a time constant of about 50 fs (Sala et al., Ref. 28). Full-dimensional quasi-classical surface-hopping simulations using the ADC(2) electronic-structure method reproduced this time constant (Xie et al., Ref. 32). While both calculations are approximate, the close mutual agreement is a strong argument in support of ab initio theory. In a couple of years, X-ray TA PP spectroscopy with considerably higher time resolution may be possible. Therefore, the authors should make an unequivocal statement whether or not their time constant of 200 fs is in contradiction with theory. Ideally, an error bar of the measurement should be given.

(2) It is known from numerous computational studies for pyrazine that the adiabatic and diabatic population probabilities are different. Depending on transition dipole moments, Franck-Condon factors and the detection technique, the measured PP signals can be closer to the adiabatic or the diabatic population probability, see, e.g., the review of Domcke and Stock, Adv. Chem. Phys. 100, 1 (1997). A statement is needed which of the two population probabilities (or neither) is measured by X-ray TA PP spectroscopy. It has been shown that the diabatic electronic populations in pyrazine exhibit weakly damped quantum beats driven by tuning modes, while the adiabatic populations are essentially structureless. With higher time resolution in the future, the vibrational quantum beats in the diabatic population probabilities may become detectable.

(3) Considering that quantum dynamics simulations at the XMCQDPT2 level and trajectory surface-

hopping simulations at the ADC(2) level are available for pyrazine, TDDFT simulations seem inadequate, the more so as it has been demonstrated in the literature (see, e.g., Sala et al., Ref. 28) that TDDFT places the energy of the B_{2g}(nπ*) state incorrectly, which has consequences for the nonadiabatic dynamics. Therefore, Sections 5 and 9 of the SI and the "computational details" in the Methods section should be skipped. They do not make any contribution to the substance of the paper. The B_{2g} state is not mentioned throughout the article. Is it conceivable that this state, which is dark in UV PP spectroscopy, contributes to X-ray TA PP signals?

(4) Section 9 of the SI also should be skipped. Spin-orbit coupling effects are not relevant on the time scale of the experiment. A single trajectory hopping to the ground state does not provide any information.

(5) The colors in Fig. 6 seem inconsistent with the figure caption. It seems that red and green are interchanged.

Reply to reviewers' comments:

We are grateful for the many helpful comments of the reviewers. We performed additional calculations and address all of the questions and concerns in the reply below. The replies are shown in blue and added material follows each reply to indicate extensive changes that are made. We trust that the revisions fully address all the concerns and that the paper is substantially improved and will be considered for publication in Nature Communications.

Reviewer #1:

Scutelnic et al report on the experimental detection of the $1A_u$ state in photoexcited pyrazine. Their method is based on ultrafast table-top HHG soft X-ray spectroscopy at the carbon K-edge with ~ 90 fs temporal resolution. The optically dark $1A_u$ state has been predicted theoretically - this would be the first experimental evidence for its population. The authors compare their challenging experimental data with ab initio TDDFT and FSSH calculations of the excited states and corresponding oscillator strengths for the core-level transitions originating in the $1s$ level.

1) Overall, this is a well-written paper and the results are presented in a clear way. The results are of high impact in the physical chemistry community. However, unfortunately, the S/N ratio in the data is lacking and my overall impression is that the assignment to the $1A_u$ state (or a mixture of $1A_u$ and $1B_{3u}$) is not very convincing.

Reply: We greatly appreciate the positive feedback about the work by the reviewer. We thank the referee for pointing out the missing uncertainties. The uncertainties have been added in the main text for the $1A_u$ lifetime: on page 2 “about 200 fs” was replaced with “in 200 ± 50 femtoseconds”, page 5 “ ~ 200 fs” was replaced with “ (200 ± 50) fs”. A new discussion of the uncertainty in the determination of the ground-state bleach has been added as Section 3 of the Supplementary Information, and this new section is recited here below. Details concerning the assignment of the $1A_u$ state are discussed in the reply 5).

Section 3 in SI: Correcting for ground state bleach

X-ray transient absorption spectra are composed of the positive absorption signals of excited states and a negative absorption signal from the ground state, as its population is depleted by the UV pump. In the experimental conditions of $\sim 30 \pm 6$ mJ \times cm $^{-2}$ pump fluence (2.5×10^{11} W \times cm $^{-2}$ intensity) and with an absorption cross section of 7 Mb at the pump wavelength of 267 nm,² up to 25% of the pyrazine molecules are excited. This estimate relies on the measurement precision of the UV pulse energy, obtained with an uncertainty of <1%, and spot size at focus, estimated to within 20%. These uncertainties propagate to an upper limit of variability in the determined excitation fraction of (25 ± 5) %. In Figure S3, the filled spectra show that the principal change in spectra with 20% and 30% add-back is most clearly revealed in the alteration of the intensity of the 285.4 eV band, which is not used in the assignment of $1B_{2u}$, $1B_{3u}$ and $1A_u$ states and does not change any of the conclusion. An UV

excitation of 23% is found to be optimal for removing the discontinuities (highlighted with vertical dashed lines in Figure S3) caused by the ground state bleach. This number is very close to the 25% estimate from the pulse energy, beam volume, and cross section.

At delays >100 fs, the pump and probe pulses are well separated in time and the estimated 23% or 25% of the excited molecules is subsequently probed. When the UV pump and X-ray probe pulses overlap in time (at -20 fs and 20 fs delays in Figure 5), the fraction of excited and probed molecules $p_{excited\&probed}$ varies depending on the pump-probe delay t , due to the temporal overlap of the UV pulse with the X-ray probe pulse, initially giving a much smaller fraction and eventually reaching the full excited population in time:

$$p_{excited\&probed}(t) = p_{exc} \cdot \int_{-\infty}^t (f_{UV} * f_{Xray})(\tau) d\tau = p_{exc} \cdot \int_{-\infty}^t IRF(\tau) d\tau = p_{exc} \cdot \frac{1 + \operatorname{erf}\left(\frac{t}{\sqrt{2} \cdot \sigma_{IRF}}\right)}{2},$$

where p_{exc} is the percent of molecules initially excited by the UV pulse (23% used in the add-back spectrum), f_{UV} and f_{Xray} are the temporal profiles of the UV and X-ray pulses (approximately gaussians), and IRF represents the instrument response function with a standard deviation $\sigma_{IRF} = (35 \pm 6) fs$ (see Figure S2). Thus, $p_{excited\&probed}(-20 fs) = 7\%$ and $p_{excited\&probed}(20 fs) = 15\%$ are applied in Figure 5.

Figure S3. Black: differential X-ray spectrum acquired at 220 fs; dashed line is reference zero line. Red: Scaled ground state. Filled blue: the sum of the transient spectrum and 23% of the ground state. Filled purple: the sum of the transient spectrum and 20% of ground state. Filled cyan: the sum of the transient spectrum and 30% of ground state.

2) One particular point of concern is the fact that the authors had to add a varying fraction of the ground-state spectrum to reconstruct the excited state NEXAFS spectra. They state that the fraction is changing over time due to a change in spatial overlap. I understand these are extremely challenging experiments, but a time-varying spatial overlap between pump and probe is really worrying..... The authors do not adequately address this problematic.

Reply: We thank the reviewer for pointing out that the stated procedure and the mechanics of the add back spectrum are unclear. The spatial overlap of the pump and probe does not vary in time. At time delays when the probe pulse precedes the pump, the percent of valence excited molecules that is probed by the X-ray pulse is dependent on the time delay and varies as determined by a formula for the pulse timing overlap developed in Section 3 of the Supplementary Information (recited above). The phrase in the footnote of main text Figure 5 “(the added ground state percent is different since the overlap of the pump and probe changes in time, see Supplementary Information).” was corrected to read: “The added ground state percent is determined from the pump fluence and absorption cross section of pyrazine at 267 nm; for -

20 and 20 fs delays the temporal overlap of the pump and probe is also taken into account, see Section 3 in Supplementary Information.”

3) From their description in the SI, it is not clear to me what criteria are applied to determine the fraction of GS spectrum that is added. Within the S/N level of the data, a range of fractions would still yield reasonable spectra (with reasonable meaning that cross sections are not negative, nor sharp peaks appear in an otherwise smoothly varying spectrum). It is not clear to me to which extent the uncertainty in ground-state contribution affects the shape of the excited-state spectrum at different time delays (and how this in turn may affect the spectral assignments and kinetic fitting).

Reply: We have developed considerable expertise over a number of publications in determining the fraction of ground state spectrum to add back in this type of X-ray time-resolved experiment. The procedure takes into account the pump beam intensity, beam size and pyrazine absorption cross section at 267 nm to determine the fraction of excited molecules, this information has been added to the main text (footnote to Figure 5) and SI (Section 3), both reproduced above. Supplementary Information has been completed with error analysis and differential spectra with largely different percents of add-back to illustrate that the bands used in assignments are not affected by this procedure. Similarly, the kinetic fitting is carried out specifically on regions of the spectrum that do not overlap with the ground state absorption and are thus independent of the fraction of excited molecules.

4) I also have some concerns about the reliability of the fitted time constants given the rather large IRF and uneven sampling along the time axis. The authors state that the fitted decay constant for the $1B_{1u}$ state is “strongly affected by the IRF”. What does this mean? Aren’t the authors fixing the IRF in the fitting and properly weighting the fit with its respective error bar? Is there additional uncertainty expected in the IRF, i.e. variations as a function of (real) time? Also, the time stepping at later delay times is not the same as at early delay times, which may affect the statistical significance of the difference in reported decay constants τ_1 and τ_2 .

Reply: We thank the reviewer for pointing out the lack of clarity. We determine the IRF in a separate experiment using the AC-Stark shift in argon and convolve the exponential decay with this IRF, but because the error bars in the data for the transient at $1B_{2u}$ could accept a large range, it results in a large uncertainty of the determined $1B_{2u}$ lifetime. The phrase on page 11 of the main text “strongly affected by the >80 fs instrumental response time (Supplementary Fig. S2).” is changed to read: “the large uncertainty is affected by the low signal-to-noise of this transient.”

The step sizes in Figures 4a,b, from which the τ_1 and τ_2 are derived, is the same: 20 fs. Figures 4c,d have more points on the slope to establish the time constants with a better precision.

5) When I look at Fig. 5 and I compare the simulated green spectra with the GSB-corrected experimental orange spectra, I am just not convinced about the agreement... In particular, the lowest-energy pre-edge peak is really off for all time delays and it is not entirely clear why (does this mean the authors aligned the theoretical and experimental spectra based on the feature around 284 eV; if yes, how does this affect the assignment of the $1A_u$ shoulder feature denoted with an asterix?). I think the authors mention the motion of the wave packet away from the FC region to be responsible for this discrepancy for the longest time delays (~220 fs). It’s not clear how this affects the earlier times though.

Reply: First, all the calculated spectra are blue shifted by a constant 10.7 eV to match the calculated spectra with the experimental spectra; this shift is determined from the alignment of the calculated ground-state spectrum with the corresponding experimental one (given in Fig. 2) and is very typical for density functional methods that address X-ray spectra, as indicated below in the revised text. Importantly, the same 10.7 eV shift is used for the calculated $1A_u$ state spectrum. To clarify this, a phrase was added on page 5 - “The computed spectrum of the ground state is blue shifted by 10.7 eV to align it with the experimental blue line.” and on page 17 - “The 10.7 eV constant shift is a systematic error of TDDFT, attributable to relaxation of the electrons caused by the core-hole and by self-interaction errors and is determined from the alignment of the calculated ground-state spectrum with the corresponding experimental one.”

Concerning the mismatch of the low-energy feature in the theoretical spectra, it is attributed to geometry effects of the molecular motion. We carried out additional calculations of the three excited-state spectra on a distribution of structures that arises from the zero-point vibrational energy (Wigner distribution) for the

ground state of pyrazine, and added a new section to the SI containing this new information, which is cited in its entirety here below. With the new calculations, the lower energy peaks of ${}^1B_{2u}$, ${}^1B_{3u}$ and 1A_u appear blue shifted by 0.3, 0.1 and 0.3 eV relative to the FC-calculated ones, respectively (see Figs. S5-S7); this already reduces the discrepancy between theory and experiment. Since these spectra for the ground-state Wigner distribution are more realistic than those at the FC geometry, we replaced the calculated spectra in Fig. 5 (added below). The separation between the lower energy ${}^1B_{2u}$ and ${}^1B_{3u}$ peaks is 0.9 eV, which is only 0.1 eV higher than the corresponding experimental value. This higher accuracy supports the assignment of the spectral signature of the 1A_u at 284.5 eV, which is shifted from the main ground state peak at 285.4 eV by -0.9 eV. The main text was completed for the new calculations accordingly, added on page 9: “which are averaged over a distribution of structures that arises from the zero-point vibrational energy (Wigner distribution) for the ground state of pyrazine” and “Section 5 of the Supplementary Information discusses geometrical effects on the low-energy ${}^1B_{2u}$ and ${}^1B_{3u}$ peaks.”; on page 11 “Although the positions of the 281.5 eV and 282.3 eV low-energy peaks are underestimated by the calculations, by 0.5 eV and 0.4 eV, respectively, the experimentally-observed 0.8 eV blue shift is well reproduced (0.9 eV).”; on page 12 “The experimental band at 284.5 eV aligns well with the modeled absorption signals at 284.4 and 284.8 eV from the 1A_u state at the FC geometry (see green curve of Fig. 5c, NTOs are in Supplementary Table S5).” this now reads “The experimental band at 284.5 eV aligns well with the modeled absorption signal at 284.85 eV from the 1A_u state for the ground-state Wigner distribution (see green curve of Fig. 5c and NTOs at the FC geometry in Supplementary Table S5).” Furthermore, we have investigated the effect of the wave packet dynamics at early times (Figs. S5 and S6). We observe that the lower energy ${}^1B_{2u}$ and ${}^1B_{3u}$ peaks shift further by 0.2 eV and 0.1 eV respectively, relative to the corresponding peaks of the Wigner-broadened spectra. This leads to a separation of 0.8 eV, which is the same value observed in the experiment. Further details on this analysis are given in Section 5.

Section 5 of SI: Geometrical effects on the calculated excited-state X-ray absorption spectra

The low-energy peaks in the experimentally extracted excited-state spectra in Figures 5a and 5b of the main article are located at 281.5 eV (-20 fs time delay) and 282.3 eV ($+20$ fs time delay). Based on the spectra calculated at the Franck-Condon (FC) geometry, the spectral signatures are assigned to the ${}^1B_{2u}$ (281.5 eV, $1s \rightarrow 1b_{1g} - \pi$) and ${}^1B_{3u}$ (282.3 eV, $1s \rightarrow 6a_g - n$) states (see Figures S5, S6 and Tables S3, S4). However, the FC-calculated peaks are centered at 280.7 eV (${}^1B_{2u}$) and 281.8 eV (${}^1B_{3u}$), leading to 0.8 eV and 0.5 eV discrepancies with the experimental values, respectively. Here we show that this mismatch decreases by accounting for geometrical effects. Figures S5 and S6 show the trajectory-averaged ${}^1B_{2u}$ (18 structures) and ${}^1B_{3u}$ spectra (28 structures) at -20 fs and $+20$ fs, respectively; -20 fs in the experiment corresponds to 140 fs in the surface hopping (SH) simulation, because the 0 fs is set to the pump maximum in the experiment. We selected the structures with minimal valence electronic mixing, because the ${}^1B_{2u}/{}^1B_{3u}/{}^1A_u$ mixing would interfere with identifying the

geometrical effect, which was not possible in the case of the 220 fs (Figure S7) time delay due to strong geometry distortion. As seen in Figure S5, the low-energy peaks are shifted to 281.2 eV (${}^1B_{2u}$) and 282.0 eV (${}^1B_{3u}$), decreasing the deviation from experiment to 0.3 eV in both cases (which is the same as the resolution of our X-ray spectrometer). Importantly, the separation of the two peaks decreases from 1.1 eV (at FC geometry) to 0.8 eV, the latter being in perfect agreement with the experimental value. We found that the peak positions are converged with respect to the number of excited-state structures utilized in the calculations. We have also calculated the averaged spectra for the ground-state Wigner distribution, resulting in 281.0 eV (${}^1B_{2u}$) and 281.9 eV (${}^1B_{3u}$) peak positions, as well as that of the 1A_u state, resulting in its 281.4 eV peak position. This shows that the geometric effect has a significant dynamic character. Note that in the simulated dynamics spectrum of Figure 6b in the main article, which is the same as the blue curve in Figure S7, geometrical effects are accounted for and the position of the low-energy peak is indeed in good agreement with the experimental one. Finally, in Figure S8, we illustrate the convergence of the simulated dynamics (trajectory-averaged) spectrum with respect to the number of utilized trajectories. This clearly shows that the simulated X-ray absorption spectrum utilizing 59 trajectories is converged.

Figure S5. Green: Computed ${}^1B_{2u}$ spectrum at the Franck-Condon geometry. Red: Average of 23 X-ray spectra for the ${}^1B_{2u}$ state obtained from the ground-state Wigner distribution. Black: Trajectory-averaged ${}^1B_{2u}$ X-ray spectrum simulated at -20 fs time delay, calculated using 18 excited-state structures. The excitation energies have been shifted by 10.7 eV.

Figure S6. Green: Computed ${}^1B_{3u}$ spectrum at the Franck-Condon geometry. Red: Average of 37 X-ray spectra for the ${}^1B_{3u}$ state obtained from the ground-state Wigner distribution. Black: Trajectory-averaged ${}^1B_{3u}$ X-ray spectrum simulated at 20 fs time delay, calculated using 28 excited-state structures. The excitation energies have been shifted by 10.7 eV.

Figure S7. Green: Computed 1A_u spectrum at the Franck-Condon geometry. Red: Average of 25 X-ray spectra for the 1A_u state obtained from the ground-state Wigner distribution. Black: Trajectory-averaged X-ray spectrum with mixed $^1B_{3u}/^1A_u$ character simulated at 220 fs time delay, calculated using 59 excited-state structures. The excitation energies have been shifted by 10.7 eV.

Figure S8. Trajectory-averaged spectrum at 220 fs constructed with 10, 20, 30, 40, 50 and 59 trajectories of the 59 trajectories of the SH simulation.

Figure 5 from main text. **Experimental differential spectra acquired at different time delays.** (a) -20 fs delay (blue), and the computed $1B_{2u}$ spectrum (green). (b) 20 fs delay (blue), and the computed $1B_{3u}$ spectrum (green). (c) 220 fs delay (blue), and the computed $1A_u$ spectrum

(green). All the computed spectra are calculated for the ground-state Wigner distribution. The brown filled spectra are the differential traces corrected for ground-state bleach. The added ground state percent is determined from the pump fluence and absorption cross section of pyrazine at 267 nm, for -20 and 20 fs delays the temporal overlap of the pump and probe is also taken into account, see Section 3 in Supplementary Information. Prevailing electron configurations of the corresponding valence excited states are sketched on the right.

6) In this context, a calculation of the ground-state spectrum is missing. This would be very useful to estimate the expected accuracy of the theory (if the GS is bad already, can we expect it to be as good/better for the ES?). Are core-hole effects in the GS and/or ES spectra expected to play a role and accounted for in the theory?

Reply: The calculated ground state spectrum was already given in Figure 2 and noted in the caption there. The major and well-known (systematic) error of TDDFT, attributable to relaxation of the electrons caused by the core-hole and by self-interaction errors, is efficiently accounted for by uniformly shifting all spectra by the value needed to align the experimental and computed XAS spectra of the ground state. In the recent reviews by Besley (WIREs <https://doi.org/10.1002/wcms.1527>) and by Rankine and Penfold (J. Phys. Chem. A, 10.1021/acs.jpca.0c11267) detailed accounts of the accuracy of the MOM-TDDFT methods for the calculation of the (TR-)XAS spectra are given.

7) The simulation of the ~220 fs spectrum using FSSH, however, shows reasonable agreement with the experiment, which adds significantly to the credibility of the conclusions. The major issues with the GSB correction, however, still make this agreement less convincing than hoped for...

Reply: As mentioned above, all spectra are shifted by the same amount, determined on the basis of the ground state. Spectra with widely varied percent in GSB correction show that the peaks used in assignment of the excited states do not change.

8) Given the very large laser fluences (30 mJ/cm) for solid state samples, the authors should carefully comment on their strategy to mitigate and check for sample damage.

Reply: The sample is rapidly flowing through the gas phase and is constantly replenished before each laser pulse as the gas leaks out of the flow cell.

9) In conclusion, given the concerns described above, I do not recommend publication in Nat Comm at this point. If the assignment of the 1Au state can be solidified by addressing these concerns, I agree that the impact would be large and publication in Nat Comm may be suitable.

Reply: We thank the referee for recognizing the impact of our research. We hope our replies have solved the referee's reservations for publication in Nat. Commun.

Reviewer #2:

X-ray transient absorption reveals the 1Au ($n\pi^*$) state of pyrazine in electronic relaxation

This manuscript reports the time-resolved X-ray absorption spectrum of pyrazine following femtosecond excitation at 267nm, thereby primarily preparing the optically bright $\pi\pi^*$ electronic state. The evolution of the electronic character of the prepared wave packet is monitored via the evolution of the X-ray absorption spectrum (XAS) and interpreted using surface hopping simulations and XAS simulated using

ab initio electronic structure methods. The authors assign the delayed rise of a prominent spectral feature to the $1A_u$ state, whose role in the electronic relaxation pathways of pyrazine has been the source of recent discussion in the literature. This work employs novel/emerging time-resolved spectroscopic techniques in conjunction with state of the art electronic structures approaches on a problem that highlights the potential of ultrafast X-ray spectroscopy to image photo-physical processes. Each of these elements make this work appropriate for publication in Nature Communications, but the key assignment of this work, that of the role of the $1A_u$ state in the excited state dynamics, is not conclusive.

Reply: We thank the referee for recognizing the perspective high impact of our work. Below we address the criticism, that we believe should remove all doubts concerning the validity of the conclusions.

The first 3 comments speak directly to the assignment of specific spectral features to the A_u state, while the remaining points address more general questions regarding the analysis of the spectrum.

1) The most significant issue concerns the high degree of overlap between the XAS from the B_{3u} and A_u states. As Figure 6 shows, given the resolution of the experiment, are these results also consistent with the wave packet being exclusively B_{3u} character? Furthermore, it is not clear, on the basis of Figures 5c and 6 alone, that the shoulder of the A_u spectrum at 284.5 eV is consistent with the dominant spectral feature in the experimental TRXAS.

Reply: The experimental data are not consistent with an excited-state wave packet being exclusively of $^1B_{3u}$ character. Only the 1A_u state can account for the intense peak at 284.5 eV, which is red-shifted from the ground-state main peak at 285.4 eV. The $^1B_{3u}$ spectrum exhibits negligible intensity near this energy (green trace in Figure 5b), as also shown in Table S4 (the nearest is the peak at 285.3 eV, which has a very small oscillator strength). Although the $^1B_{3u}/^1A_u$ signals in the low-energy region do overlap, the 284.5 eV spectral feature cannot be accounted for unless the 1A_u state is included in the simulations. In panels (c)-(e) of Figure 6 we now show better an example of the spectra for the ‘pure $^1B_{3u}$ ’, ‘pure 1A_u ’ and ‘mixed $^1B_{3u}/^1A_u$ ’ states sampled at the same time. The example illustrates that essentially all trajectories show the following at all times: $^1B_{3u}$ alone cannot be responsible for the absorption peak at 284.5, labeled with a star * – its characteristic peak lies above this energy value. Figure 6 (given below) has been reformatted to facilitate the comparison with the experimental 220 fs. On page 13 of main text a sentence “The remaining 56 spectra show analogous characteristics, apart from the two trajectories which remain in the $^1B_{2u}$ state” was added.

Figure 6. NEXAFS spectra of selected geometries from the surface hopping simulation. Brown area is the differential spectrum at 220 fs corrected for ground-state bleach (see Figure 5c). Red curve is the spectrum averaged over all the geometries of the 59 trajectories of the FSSH simulation at 220 fs. Blue, green and purple lines are calculated for different geometries extracted from the trajectories at 220 fs. A NTO pair in each panel shows the dominant transition character at the chosen molecular geometry. Asterisks indicate the transitions characteristic of the 1A_u state.

2) It does not appear that the peak at 284.5eV decays even after 200ps. The authors attribute this behavior to the rise in vibrationally excited ground electronic state population. While this assignment for the long time (> 20ps) is reasonable, without an observable decay how is it possible to separate excited signal from

ground state signal? Therefore, the key peak used in the assignment overlaps not only with a different excited state, but also the ground state.

Reply: Even though there is no clear decay of the 284.5 eV signal, the 1A_u and hot ground state are well separated in time: internal conversion to the ground state takes place on a picosecond timescale, as shown in numerous experimental works, thus, the hot ground-state signal does not interfere with the analysis and appearance of the sub-ps timescale dynamics. This is discussed in detail in Section 9 of the SI. The main text was modified on page 15 with a reference to this section: “However, the internal conversion to the hot ground state in the first 200 fs is excluded (former Section 10, current Section 9 of Supplementary Information).”

3) In Figure 5c, the peak at 284.5 is seemingly assigned to the a peak in the A_u spectrum at the FC region. On the basis of the discussion, is this reasonable? The trajectory simulations appear to indicate that the structural relaxation results in a shift of the “Au” spectrum. The trajectory-averaged signal shows only a small shoulder at this energy.

Reply: Among the spectra calculated for the three states, only the one for the 1A_u state is consistent with the 284.5 eV feature. The structural relaxation in the trajectory simulation leads to a significant shift of the low energy band from the peak in FC spectrum, while the 284.5 eV band remains at nearly the same energy. Even though molecular geometrical changes may affect the spectrum, without the contribution from the 1A_u excited state, the averaged signal does not show the shoulder at 284.5 eV. Following a similar comment from Reviewer #1, Section 5 is added to the SI (entirely cited in the reply to the Reviewer #1) to clarify the geometrical effects on the calculated excited-state X-ray spectra. The shoulder at 284.5 eV in the trajectory simulation is assigned exclusively to the 1A_u state, but the 286 eV peak is due to both $^1B_{3u}$ and 1A_u states (see Figures 6c,d), giving reason to its high intensity.

4) The authors’ analysis that yields a result 23% of the sample excited via a single photon process is quite high. It is difficult to excite more than 10% of a sample before multi-photon absorption processes occur. The fact that the IP of pyrazine is almost exactly $2 \times 4.67 \text{ eV}$ ($\sim 9.3 \text{ eV}$) adds to this concern. Did the authors confirm pyrazine cation is not contributing to the TRXAS?

Reply: The possibility of cation formation was considered in detail during the experimental work and in the preparation of the original manuscript, as we have considerable experience with this from other studies. There are a few factors that need to be considered. First, the absorption cross section of pyrazine is very large at this UV wavelength, which makes it easy to excite with one photon compared to a much less efficient two-photon process. Second, if a two-photon process would occur, the formed ion would be in its ground state with nearly zero internal energy and would not be able to undergo dynamics on the 200 fs timescales, thus the ionic features would appear abruptly on the duration of the instrument response function, as in the case of the benzene radical cation (ref. 39 in the main text). Instead, the 284.5 eV band assigned to 1A_u has a rise time more than two times longer than the IRF. Third, we tested the one-photon dependence of the 284.5 eV signal on the UV power (Figures S12 and S13). The Supplementary Information was modified on page 18 with additional relevant data on this subject (see below).

Addition to SI on page 18:

As the absorption cross section of pyrazine is very high ($7 \text{ Mb at } 267 \text{ nm}$)² the one-photon excitation is more efficient than the two-photon ionization. Moreover, if a two-photon process would occur, the formed ion would be in its ground state with nearly zero internal energy (pyrazine ionization potential 9.0 eV ,¹³ two 267 nm photons carry about 9.3 eV) and would not be able to undergo dynamics on the 200 fs timescales, thus the ionic features would appear on the duration of the IRF as in the case of the benzene radical cation.¹⁴ Instead, the 284.5 eV band assigned to the 1A_u state has a time constant more than twice as long as the IRF. To verify that the UV excitation in pyrazine is indeed due to one photon, the X-ray spectra were recorded at different UV powers (Figure S12). Lower pump power results in a weaker signal, while the overall shape of the spectrum does not change significantly, in particular the ratio of the bands peaking at 282.3 and 284.5 eV is nearly the same.

Figure S12. Differential absorption spectra acquired at 3 ps delay. Shaded area represents one standard deviation of 128 measurements.

5) Regarding Figure 5: why was the trajectory averaging procedure only performed for the 220fs spectrum, this analysis would also be relevant for showing vibrational/nuclear dynamics on the B_{3u} state?

Reply: We have now extended our trajectory averaging analysis to consider the wave packet dynamics of the ¹B_{2u} and ¹B_{3u} states at early times (Figs. S5 and S6 in Section 5 of SI, reproduced in the reply to the Referee #1). Trajectory simulations show that the lower energy peak of the ¹B_{2u} state shifts to higher energy by 0.5 eV and the features of the ¹B_{3u} by 0.2 eV, decreasing the discrepancy with experiment. The separation between the two peaks is 0.8 eV, which is the same value as in the experiment.

6) A surface hopping simulation employing only 59 trajectories (with this many degrees of freedom and this many electronic states) is likely not converged (eg. wrt to electronic populations). While the authors don't lean on these results too hard, this would affect both the fraction/prevalence of Au state present in the simulation, but perhaps more importantly, how energetically broad the expected spectral signature would be at, for example 220fs.

Reply: By constructing the trajectory-averaged spectrum with 10, 20, 30, 40, 50 and 59 trajectories, we confirmed that the surface hopping simulation employing 59 trajectories is converged (Figure S8 in the SI, also reproduced below). A sentence explaining this has been added to the Computational Details section of the main manuscript: "The convergence was tested by computing the trajectory-averaged spectrum with 10, 20, 30, 40, and 50 of the 59 trajectories, which confirmed thereby that the surface hopping simulation employing 59 trajectories was converged (see Supplementary Fig. S8)."

Figure S8. Trajectory-averaged spectrum at 220 fs constructed with 10, 20, 30, 40, 50 and 59 trajectories of the 59 trajectories of the SH simulation.

In summary, this work is of high quality and employs powerful new ultrafast spectroscopy and high levels of ab initio computation to investigate a benchmark photophysical system. However, the results and analysis are unable to definitively assign the contribution of an elusive optically dark electronic state to the wave packet dynamics, in the opinion of this referee, which makes it more suitable for a more specialized journal.

Reviewer #3:

In this work, the femtosecond relaxation dynamics of the valence excited states of the pyrazine molecule in

the gas phase was investigated with X-ray transient absorption (TA) pump-probe (PP) spectroscopy. While pyrazine as such is not a particularly exciting chromophore, a wealth of experimental and theoretical data on the femtosecond relaxation dynamics of the valence excited states of pyrazine are available. The interpretation of the signals of the relatively new X-ray TA PP technique can thus be scrutinized in comparison with well-established knowledge. It is very interesting that the time-dependent population of the Au(np π^*) state can be measured with X-ray TA PP spectroscopy, because the population dynamics of this dark state cannot be detected with conventional UV TA PP spectroscopy. The present work is of broad interest for the highly active field of ultrafast X-ray spectroscopy with novel light sources and is therefore recommended for publication in Nature Communications.

Reply: We thank the referee for the positive evaluation and for his/her recommending publication.

A few issues should be clarified in a revision of the manuscript.

1) The authors report a time constant of 200 fs for the transfer of population from the bright B_{2u}(p π^*) state to the dark Au(np π^*) state. Quantum dynamics calculations for a nine-dimensional model derived from XMCQDPT2 ab initio calculations predicted a time constant of about 50 fs (Sala et al., Ref. 28). Full-dimensional quasi-classical surface-hopping simulations using the ADC(2) electronic-structure method reproduced this time constant (Xie et al., Ref. 32). While both calculations are approximate, the close mutual agreement is a strong argument in support of ab initio theory. In a couple of years, X-ray TA PP spectroscopy with considerably higher time resolution may be possible. Therefore, the authors should make an unequivocal statement whether or not their time constant of 200 fs is in contradiction with theory. Ideally, an error bar of the measurement should be given.

Reply: The uncertainty of the transient (200 \pm 50) fs has now been added throughout the main text (the error bars were given earlier in Figure 4c).

On the one hand, our experimental results indicate a time constant of (200 \pm 50) fs, which is in contradiction with the theoretical results of Sala et al (ref. 28) that predict a time constant of 50 fs. On the other hand, the growth of the 284.5 eV peak with lifetime of (200 \pm 50) fs is qualitatively consistent with the growth of population in the ¹A_u state in later work by Sala et al. (Ref. 29). Moreover, it would appear from the results of Ref. 29 that the theoretical estimates of the time constant depended on the pulse duration in their simulations, and so does their experimental value. It does not seem prudent to assess the theoretical work of Sala, but we have clarified this point in the main text with the following sentence: "A slightly delayed increase of the population in the ¹A_u state during the ultrafast relaxation (Figure 4c) is qualitatively consistent with the nuclear dynamics simulation of Sala et al.²⁹,"

2) It is known from numerous computational studies for pyrazine that the adiabatic and diabatic population probabilities are different. Depending on transition dipole moments, Franck-Condon factors and the detection technique, the measured PP signals can be closer to the adiabatic or the diabatic population probability, see, e.g., the review of Domcke and Stock, Adv. Chem. Phys. 100, 1 (1997). A statement is needed which of the two population probabilities (or neither) is measured by X-ray TA PP spectroscopy. It has been shown that the diabatic electronic populations in pyrazine exhibit weakly damped quantum beats driven by tuning modes, while the adiabatic populations are essentially structureless. With higher time resolution in the future, the vibrational quantum beats in the diabatic population probabilities may become detectable.

Reply: X-ray TA PP spectroscopy probes the population of the diabatic states, in the sense that it is sensitive to the electronic character. For this reason, we have based our discussion on the Franck-Condon geometry, where the three electronic states belong to different irreducible representations and are therefore pure/diabatic states. However, during the dynamics the structure is distorted and the D_{2h} symmetry is lost, and therefore the diabatic states mix. This mixing is already discussed on page 13 (Fig 6). We agree with the referee on the future possibility of observing the vibrational quantum beats.

3) Considering that quantum dynamics simulations at the XMCQDPT2 level and trajectory surface-hopping simulations at the ADC(2) level are available for pyrazine, TDDFT simulations seem inadequate, the more so as it has been demonstrated in the literature (see, e.g., Sala et al., Ref. 28) that TDDFT places the energy of the B_{2g}(np π^*) state incorrectly, which has consequences for the nonadiabatic dynamics.

Therefore, Sections 5 and 9 of the SI and the “computational details” in the Methods section should be skipped. They do not make any contribution to the substance of the paper. The B_{2g} state is not mentioned throughout the article. Is it conceivable that this state, which is dark in UV PP spectroscopy, contributes to X-ray TA PP signals?

Reply: The ¹B_{2g} state is not included in our analysis, especially since it is significantly higher in energy than the observed states. Therefore, we also do not consider it problematic that TDDFT may be imprecise in estimating the energy. In the study of Sala et al., Ref. 28 the ¹B_{2g} state was shown to be insignificant in the deexcitation. Following the referee’s suggestion, we have omitted the old Section 5 of the SI. We have chosen to keep the Computational Details that ensure reproducibility of the results. It is possible that important details could easily be overlooked for instance as to how we matched theoretical and experimental spectra, which is described in the Computational Details. As for the old Section 9, we refer to our reply to the next comment below.

4) Section 9 of the SI also should be skipped. Spin-orbit coupling effects are not relevant on the time scale of the experiment. A single trajectory hopping to the ground state does not provide any information.

The aim of the single trajectory hopping was to collect representative geometries of the vibrationally-hot ground state and not to trace intersystem crossing. This was used to consider the long-time behavior of the observed spectrum (page 15 in the main manuscript). Thus, we have chosen to retain the section 9 as the new Section 8 of the SI, but we have changed its title (*Simulation of the hot ground-state spectrum at long delay*) to avoid misunderstanding of its purpose.

5) The colors in Fig. 6 seem inconsistent with the figure caption. It seems that red and green are interchanged.

Reply: We thank the referee for noticing this error, and it has been rectified.

REVIEWERS' COMMENTS

Reviewer #1 (Remarks to the Author):

The comments have been adequately addressed by the authors. The reviewer appreciates all the efforts invested to improve the manuscript. I still think the agreement between theory and experiment is not 150% convincing, especially also given the fact that the entire conclusion of seeing the optically dark 1Au state is based on a weak shoulder feature around 284.5 eV, the strength of which varies depending on the amount of GS that is added to the transient spectrum (which cannot be determined very accurately). However, given all the extra data provided by the authors and reading some of the replies to the other reviewers, I do think the conclusion is more credible than it was for the first manuscript. My balance just slightly tips over into the direction of recommendation for publication in Nat Comm. As one of the other reviewers points out, near-future improvements in technical capabilities (S/N and time resolution) will probably/hopefully/beautifully nail this down...

Reviewer #2 (Remarks to the Author):

X-ray transient absorption reveals the 1Au (nn*) state of pyrazine in electronic relaxation

1. "Only the 1 Au state can account for the intense peak at 284.5 eV, which is red-shifted from the ground-state main peak at 285.4 eV".

From the Figure 6, it can be seen that the B3u state has a shoulder off the main peak that is red-shifted by $\sim 0.7-0.9$ eV. When the B3u spectrum is shifted to match the absorption maximum observed in the experiment, this shoulder is in the correct location as the purported signature of the Au state – even if the intensity (based on a TDDFT computation) would seem a little low. This raises the question: would the authors expect to the "Au" spectrum to also require a similar shift? And if so, does the assignment still stand?

Addressing this question more broadly: there will be an inherent error (likely ~ 0.5 eV, depending on electronic structures involved) associated with the electronic structure method used to simulate the spectra and that serves as the primary basis for the assignment. Given the degree of overlap between the B3u and Au spectra, the authors are demanding a lot from the simulations/electronic structure methods to parse small differences in between the component electronic states in the wave packet. Even with the revisions the statement that "only" the Au can account for the peak at 284.5 eV is overly strong.

However, the addition of Figures S5-S7 to SI are powerful arguments in favor of the authors assignment. Specifically, it would appear that the B3u initial state XAS does not appear to have a pronounced geometry dependence, while the Au signal may increase in the spectral region of interest following nuclear relaxation. These figures are perhaps more evocative than the current Figure 6 in the manuscript.

2. "Even though there is no clear decay of the 284.5 eV signal, the 1 Au and hot ground state are well separated in time: internal conversion to the ground state takes place on a picosecond timescale, as shown in numerous experimental work..."

The intent of the previous comment from the original report was to note that a transient signal would support the assignment of the spectral feature to an excited electronic state. As the authors correctly state, the fact this feature does not decay does not preclude it's origin at short times to a transient

excited state. However, the fact the key spectral feature of this work, used to identify the contribution of a 'debated' (though not widely) excited state state in the dynamics, does not display the expected signature of a transient excited state feature, is unfortunate.

While the current work cannot make a definitive assignment of the "Au" state in the dynamics (the use of the adiabatic electronic state basis for the dynamics also complicates this type of assignment), it provides compelling evidence for a particular electronic relaxation pathway using a novel experimental technique.

The authors revisions have made the manuscript suitable for publication in Nature Communications.

Reviewer #3 (Remarks to the Author):

The authors responded satisfactorily to my comments, which addressed minor issues.

The manuscript is recommended for publication in Nature Communications.

REVIEWERS' COMMENTS

Reviewer #1 (Remarks to the Author):

The comments have been adequately addressed by the authors. The reviewer appreciates all the efforts invested to improve the manuscript. I still think the agreement between theory and experiment is not 150% convincing, especially also given the fact that the entire conclusion of seeing the optically dark $1A_u$ state is based on a weak shoulder feature around 284.5 eV, the strength of which varies depending on the amount of GS that is added to the transient spectrum (which cannot be determined very accurately). However, given all the extra data provided by the authors and reading some of the replies to the other reviewers, I do think the conclusion is more credible than it was for the first manuscript. My balance just slightly tips over into the direction of recommendation for publication in Nat Comm. As one of the other reviewers points out, near-future improvements in technical capabilities (S/N and time resolution) will probably/hopefully/beautifully nail this down...

We thank the reviewer for his/her recommendation for publication. We would like to underline that the peak at 284.5 eV is separated from the first peak in the ground-state spectrum by -0.9 eV. Such a considerable separation makes the shoulder structure noticeable even when the ground-state spectrum weighted by 0.3 is added back to the transient spectrum (see Supplementary Figure 3). However, as it is true that the key spectral feature is a shoulder of moderate intensity, we understand the referee's note of caution and suggestion to avoid too strong statements. We have modified the text in the Introduction ("The combined results of experiment and theory provide strong evidence for" is replaced with "The evidence obtained from experiment and theory points towards") and Discussion ("The present work provides direct evidence" is replaced with "The evidence in the present work supports the substantial role of the $1A_u$ state") to soften the tone when presenting our assignments, and look forward to near-future improvements in technical capabilities to further confirm our conclusions.

Reviewer #2 (Remarks to the Author):

X-ray transient absorption reveals the $1A_u$ ($n\pi^*$) state of pyrazine in electronic relaxation

1. "Only the 1 Au state can account for the intense peak at 284.5 eV, which is red-shifted from the ground-state main peak at 285.4 eV".

From the Figure 6, it can be seen that the B_{3u} state has a shoulder off the main peak that is red-shifted by $\sim 0.7\text{--}0.9\text{ eV}$. When the B_{3u} spectrum is shifted to match the absorption maximum observed in the experiment, this shoulder is in the correct location as the purported signature of the Au state – even if the intensity (based on a TDDFT computation) would seem a little low. This raises the question: would the authors expect to the " Au " spectrum to also require a similar shift? And if so, does the assignment still stand?

Throughout this work, we apply one unique shift of +10.7 eV to *all* spectra. Its value was chosen to match the computed with the experimental ground-state spectra, as described in the subsection "Computational details" and it represents a measure of the systematic errors of TDDFT. The assignments are based on the relative peak positions reproduced in the calculations. As emphasized in the response to Reviewer #1, the peak at 284.5 eV is separated from the first peak in the ground-state spectrum at 285.4 eV by -0.9 eV . The ${}^1B_{3u}$ shoulder in Figure 6c, which is around 285.3 eV, clearly cannot account for a separation of -0.9 eV . We agree with the referee that it would be unjustified to shift the X-ray spectra for the excited states by individual and arbitrarily chosen values.

Addressing this question more broadly: there will be an inherent error (likely $\sim 0.5\text{ eV}$, depending on electronic structures involved) associated with the electronic structure method used to simulate the spectra and that serves as the primary basis for the assignment. Given the degree of overlap between the B_{3u} and Au spectra, the authors are demanding a lot from the simulations/electronic structure methods to parse small differences in between the component electronic states in the wave packet. Even with the revisions the statement that "only" the Au can account for the peak at 284.5 eV is overly strong.

The ${}^1B_{3u}$ and 1A_u spectra do not overlap at 284.5 eV. The assignments are based on the relative peak positions. Specifically, the 0.9 eV red shift of the key spectral feature at 284.5 eV is certainly larger than the error associated with the electronic structure method (TDDFT) for this energy difference. However, as also stated in the response to the first reviewer, we recognize that our assignment is based on a moderately intense

signal, and have therefore modified the text to adopt a softer tone while presenting our proposed assignment (the last sentence in Introduction “The combined results of experiment and theory provide strong evidence for” is replaced with “The evidence obtained from experiment and theory points towards” and in Discussion “The present work provides direct evidence” is replaced with “The evidence in the present work supports the substantial role of the 1A_u state”).

However, the addition of Figures S5-S7 to SI are powerful arguments in favor of the authors assignment. Specifically, it would appear that the B3u initial state XAS does not appear to have a pronounced geometry dependence, while the Au signal may increase in the spectral region of interest following nuclear relaxation. These figures are perhaps more evocative than the current Figure 6 in the manuscript.

We completely agree, and we have now added a reference to Supplementary Figs. 6,7 and related text in the main manuscript on page 13.

2. "Even though there is no clear decay of the 284.5 eV signal, the 1 Au and hot ground state are well separated in time: internal conversion to the ground state takes place on a picosecond timescale, as shown in numerous experimental work..."

The intent of the previous comment from the original report was to note that a transient signal would support the assignment of the spectral feature to an excited electronic state. As the authors correctly state, the fact this feature does not decay does not preclude it's origin at short times to a transient excited state. However, the fact the key spectral feature of this work, used to identify the contribution of a 'debated' (though not widely) excited state state in the dynamics, does not display the expected signature of a transient excited state feature, is unfortunate.

This is indeed unfortunate but overlapping signals are inherently present in time-resolved spectroscopy. Our assignment at long time delays (>20 ps) is supported both by our theoretical analysis (Supplementary Figure 11) and the timescale of internal conversion to the ground state known from the literature.

While the current work cannot make a definitive assignment of the "Au" state in the dynamics (the use of the adiabatic electronic state basis for the dynamics also complicates this type of assignment), it provides compelling evidence for a particular

electronic relaxation pathway using a novel experimental technique.

The authors revisions have made the manuscript suitable for publication in Nature Communications.

We thank the reviewer for the recommendation.

Reviewer #3 (Remarks to the Author):

The authors responded satisfactorily to my comments, which addressed minor issues.

The manuscript is recommended for publication in Nature Communications.

We thank the reviewer for the wholehearted approval.